# A Label-Free Cell Sorting Approach to Highlight the Impact of Intratumoral Cellular Heterogeneity and Cancer Stem Cells on Response to Therapies

**DOI:** 10.3390/cells11152264

**Published:** 2022-07-22

**Authors:** Céline Hervieu, Mireille Verdier, Elodie Barthout, Gaëlle Bégaud, Niki Christou, Magali Sage, Julie Pannequin, Serge Battu, Muriel Mathonnet

**Affiliations:** 1UMR INSERM 1308-CAPTuR “Control of Cell Activation in Tumor Progression and Therapeutic Resistance”, Ω-Health Institute, Faculty of Medicine, University of Limoges, 87025 Limoges, France; celine.hervieu@unilim.fr (C.H.); mireille.verdier@unilim.fr (M.V.); elodie.barthout@unilim.fr (E.B.); gaelle.begaud@unilim.fr (G.B.); niki.christou@unilim.fr (N.C.); serge.battu@unilim.fr (S.B.); 2Department of General, Endocrine and Digestive Surgery, University Hospital of Limoges, 87025 Limoges, France; 3BISCEm US42 INSERM-UAR 2015 CNRS “Integrative Biology Health Chemistry Environment”, Ω-Health Institute, 87025 Limoges, France; magali.sage@unilim.fr; 4UMR 5203 CNRS-INSERM, Institute of Functional Genomics, University of Montpellier, 34094 Montpellier, France; julie.pannequin@igf.cnrs.fr

**Keywords:** cancer stem cells, colorectal cancer, label-free cell sorting, chemoresistance, intratumoral cellular heterogeneity

## Abstract

Cancer stem cells play a crucial role in tumor initiation, metastasis, and resistance to treatment. Cellular heterogeneity and plasticity complicate the isolation of cancer stem cells. The impact of intra-tumor cellular heterogeneity using a label-free approach remains understudied in the context of treatment resistance. Here, we use the sedimentation field–flow fractionation technique to separate, without labeling, cell subpopulations of colorectal cancer cell lines and primary cultures according to their biophysical properties. One of the three sorted cell subpopulations exhibits characteristics of cancer stem cells, including high tumorigenicity in vivo and a higher frequency of tumor-initiating cells compared to the other subpopulations. Due to its chemoresistance, two- and three-dimensional in vitro chemosensitivity assays highlight the therapeutic relevance of this cancer stem cell subpopulation. Thus, our results reveal the major implication of intra-tumor cellular heterogeneity, including cancer stem cells in treatment resistance, thanks to our label-free cell sorting approach. This approach enables—by breaking down the tumor—the study the individualized response of each sorted tumor cell subpopulation and to identify chemoresistance, thus offering new perspectives for personalized therapy.

## 1. Introduction

Despite therapeutic advances, colorectal cancer (CRC) remains a major cause of mortality worldwide [1]. Early diagnosis plays a crucial role in the survival of CRC patients. At diagnosis, patients with localized CRC account for 37% of cases, while those with metastatic stage account for 22%, according to the Surveillance, Epidemiology, and End Results (SEER) program [1]. Importantly, 5-year survival is strongly related to CRC stage. Indeed, the 5-year survival is around 90% for localized CRC, while it decreases sharply for metastatic CRC with only 14% survival. Thus, a better understanding of the cells responsible for CRC progression, metastasis, and treatment resistance is required. Although new treatment options are available, chemotherapy remains a standard therapy for CRC after surgery [2]. Treatment protocols for patients with localized or advanced CRC include 5-fluorouracil(5-FU)-based chemotherapies—such as FOLFOX (5-FU, leucovorin, and oxaliplatin), FOLFIRI (5-FU, leucovorin, and irinotecan), or FOLFIRINOX (5-FU, leucovorin, oxaliplatin and irinotecan) [3,4]. Therapeutic resistance and relapse are responsible for decreased 5-year survival in advanced CRC. One explanation for therapy failure is the presence of a minor cell subpopulation, cancer stem cells (CSCs), which are resistant to conventional therapies and therefore likely to cause tumor relapse [5]. CSCs are a small population of cancer cells endowed with self-renewal, multi-lineage differentiation, and tumor-initiating capacity in immunocompromised mice [6,7]. The main challenge in studying CSC is their identification and isolation. Surface markers were originally and for a long time used to identify CSCs. However, due to CSC plasticity and the shared expression of some markers by intestinal stem cells or cancer cells, the use of markers is no longer sufficient to define a CSC [8,9]. Currently, functional characteristics such as tumorigenic potential, relative quiescence, and chemoresistance are more often used to identify CSCs. The CSC state is subject to cellular plasticity and is inherent to intratumoral cellular heterogeneity. A few years ago, our laboratory adapted the sedimentation field–flow fractionation (SdFFF) as a cell sorting technique [10]. SdFFF is a gentle, non-invasive method that does not require cell labeling or fixation [10]. Cell separation by SdFFF depends on the differential elution of cells subjected to both the action of a parabolic profile generated by the mobile phase in the separation channel and a multi-gravitational external field generated by rotation of this channel [9]. The separation of cells is based on their biophysical properties, such as size and density, using the hyperlayer elution mode of SdFFF [11].

In previous work, our laboratory used SdFFF as a tool to isolate cell subpopulations independently of surface marker expression from CRC cell lines [10]. Results showed that one of the cell subpopulations sorted by SdFFF exhibited CSC-like features in two early-stage CRC cell lines [10,12]. However, the ability of this cell subpopulation to initiate tumors in mice, which is the gold standard for defining a CSC, remains to be determined. In our study, we therefore further characterized the phenotypic and functional characterization of the sorted cell subpopulations and assessed their ability to generate tumors in vivo. Furthermore, because of the involvement of CSCs in tumor progression and metastasis, we performed these experiments using late-stage CRC cell lines, in addition to early-stage ones, as well as primary cultures. Finally, as CSCs play a key role in resistance to conventional therapies, we explored the response of SdFFF-sorted cell subpopulations to chemotherapies. Thus, our study aimed to highlight the impact of intratumoral cellular heterogeneity on treatment resistance in order to better understand the mechanisms involved and to provide leads for more personalized therapies. In agreement with previous results, we reported here that SdFFF-sorted cell subpopulations have distinct phenotypic and functional characteristics [10]. Nevertheless, in our study, the expression of CSC markers did not necessarily correlate with their functional characteristics. The characterization results showed that a cell subpopulation exhibited the functional features of CSCs, including in vivo tumorigenicity and resistance to conventional therapies. This observation prevailed for both early- and late-stage cell lines as well as for primary cultures. Therefore, we highlighted a label-free cell sorting approach to study the individualized response of each tumor cell subpopulation in order to understand the complexity of intra-tumor cell heterogeneity and to identify chemoresistance. Ultimately, this approach could provide valuable information from CRC patient samples, offering new perspectives for more personalized therapy.

## 2. Materials and Methods

### 2.1. Cell Cultures

Four colorectal cancer cell lines were used. The two early-stage CRC cell lines, labeled “primary tumors” in the graphs, WiDr (CCL-218 ™) and SW480 (CCL-228 ™); as well as the two metastatic cell lines, labeled “metastasis” in the graphs, SW620 (CCL-227 ™) and T84 (CCL-248 ™), were obtained from the American Type Culture Collection (ATCC/LGC Promochem, Molsheim, France). SW480 and SW620 were established from the same patient, SW480 was from the patient’s primary tumor while SW620 was from lymph node metastases. The T84 cell line was derived from lung metastasis of a CRC patient. Cells were maintained at 37  °C under humidified 5% CO_2_ in MEM medium (no. 31095-029, Gibco ™, Thermo Fisher, France) for WiDr and RPMI GlutaMAX medium (no. 61870-010, Gibco ™, Thermo Fisher, France) for SW480, SW620, and T84 cell lines, both supplemented with 10% fetal bovine serum (FBS), 1 mM sodium pyruvate (no. 11360-039, Gibco ™, Thermo Fisher, France), 100 IU/mL penicillin and 100 mg/mL streptomycin (no. 15140-122, Gibco ™, Thermo Fisher, France), and only for WiDr 1% non-essential amino acids (no. 11140-035, Gibco ™, Thermo Fisher, France) [13].

Two primary cultures from CRC patients (CPP14 and CPP35) were obtained from the Institute of Functional Genomics (Univ. Montpellier, CNRS, INSERM, Montpellier, France), after informed consent of patients (Material Transfer Agreement CNRS 190287). The primary culture CPP14, labeled “early-stage tumor” in the graphs, was derived from a patient with T2N0M0 CRC—i.e., early stage (stage I)—whereas CPP35, labeled “tumor-invaded peritoneum” in the graphs, was derived from a patient with T4aN0M0 CRC—i.e., a stage at which the tumor had invaded the peritoneum (stage IIB)—as indicated in Table 1 [14]. Both primary cultures are treatment-naive. The culture conditions for these primary cultures were DMEM GlutaMAX medium (no. 61965059, Gibco ™, Thermo Fisher, France) supplemented with 10% FBS, 100 IU/mL penicillin and 100 mg/mL streptomycin, at 37 °C under a humidified atmosphere of 5% CO_2_.

### 2.2. SdFFF Cell Sorting

The SdFFF technique used enables sorting of cell subpopulations as described and schematized previously [9,10]. Cell-solid phase interactions in SdFFF are limited due to the use of a ribbon-like empty channel with no stationary phase and a size/density-based separation mechanism through the hyperlayer elution mode. SdFFF parameters used during cell sorting are field (units of gravity, g), flow rate of the mobile phase which is sterile phosphate-buffered saline (PBS, pH 7.4, no. 14190-094, Gibco ™, Thermo Fisher, France) and the rotation speed of the channel (revolutions per minute, rpm) which is related to the field. Adjustment steps were performed in order to choose the SdFFF parameters that allow an optimal separation between the void volume peak and the peak containing the cells, and to define the best elution conditions for all cell lines and primary cultures used. Elution conditions are summarized in Table 2, with a check of these parameters before and during each cell sorting.

The injected volume of the cell suspension, 100 µL, as well as the detection wavelength, 254 nm, are common for each cell line and primary culture. A cleaning and decontamination procedure was performed at the end of each cell sorting [15]. Once sorted, cells can be recultured in vitro and characterized, as no cell fixation or labeling is required for SdFFF. In order to perform experiments with the sorted cells, successive injections and collections of the same cell suspension (>10) are required to obtain a sufficient quantity of cells.

### 2.3. CSC Marker Expression

After cell sorting by SdFFF, cell subpopulation concentrations were standardized to the same amount of cells in each condition. Anti-CD44, anti-LGR5, anti-CD133/1, and viability marker antibodies were added to the cells and incubated for 30 min at 4 °C and in the dark, antibody references are summarized in Appendix A. The viability marker was used to exclude nonviable cells. Cells were then fixed in 4% paraformaldehyde (PFA, no. 10231622, Fischer Scientific) for 10 min at room temperature and permeabilized with Perm Buffer III (no. 558050, BD Phosflow ™, BD Biosciences, France) for 30 min at 4 °C. Next, antibodies recognizing the intracellular marker BMI-1 were added and incubated for 30 min at 4 °C in the dark (reference in Appendix A). As reference controls, anti-IgG2bκ FITC, anti-IgG2bκ PE-Vio 770, anti-IgG1 PE-Vio 615, and mouse anti-IgG1κ PE isotype controls were used under the same conditions and concentrations to ensure specific recognition of our antibodies of interest and set gates (references in Appendix A). Samples were analyzed by the CytoFlex LX and data analysis using Kaluza software v2.1 (Beckman Coulter, Indianapolis, IN, USA)

### 2.4. Cell Cycle Analysis

Cell concentrations of SdFFF-sorted subpopulations were standardized in each condition. After a centrifugation step, the cells were resuspended with cold PBS, fixed with cold 96% ethanol added slowly and under shaking, and then placed at −20 °C. After a few minutes at room temperature, the cells were washed and then resuspended with PBS and RNAse A (no. R6148, Sigma-Aldrich, France) for 20 min at room temperature. Next, propidium iodide was added 15 min before acquisition on the FACS Calibur. Data analysis was performed with ModFit LT^TM^ software v5.0 (Verity Software House, Topsham, ME, USA)

### 2.5. Clonogenic Assay: Soft Agar Assay

This assay was based on the use of two gels, the first is a 0.5% agar gel that prevents the cells from adhering to the culture plate and the second was a 0.7% agarose gel containing the cells. The agar (no. A7002, Sigma-Aldrich, France) and agarose (no. A9539, Sigma-Aldrich, France) solutions were prepared upstream with sterile PBS and autoclaved to limit potential contamination. Agar gel was prepared in advance and plated into wells of 24-well plates at room temperature and under the culture hood. Once the agar gel solidified, the agarose gel was heated and then gently mixed with cell subpopulations sorted by SdFFF to have a cell concentration of 1 × 10^3^ cells per well (12 wells/condition). As soon as the second gel solidified, culture medium was added on top to prevent evaporation and 24-well culture plates were incubated at 37 °C under a humidified atmosphere of 5% CO_2_ for 30 days. Four weeks later, the formed colonies were fixed in 4% PFA (no. 10231622, Fischer Scientific, France) for 15 min and then stained with 0.1% crystal violet. Wells of the 24-well plates used were captured with the Leica DFC300 FX Digital Color Camera to allow colony quantification and analyzed by ImageJ software v1.53e (National Institutes of Health, Bethesda, MD, USA)

### 2.6. In Vivo Tumor Initiation Assay

SdFFF-sorted cell subpopulations of the WiDr cell line were injected subcutaneously into nude mice (Hsd:Athymic Nude-Foxn1nu nu/nu, six weeks old, female, five mice per group) in decreasing amounts of cells (1000, 500, and 100) in Matrigel (no. 356237, Corning)–MEM medium. Mice weight and tumor size were measured three times a week for 7 weeks. After 50 days, mice were sacrificed and tumors were collected. The number of tumor-bearing mice with a tumor volume greater than 100 mm^3^ was counted. The online software Extreme limiting dilution analysis (ELDA) was used from the in vivo results to determine the frequency of tumor occurrence and thus tumor initiating cells frequency (https://bioinf.wehi.edu.au/software/elda/, accessed on 28 January 2022) [16]. This animal experimentation protocol has been approved by the Ethics Committee for Animal Experimentation no. 33 and by the French Ministry of Higher Education, Research and Innovation (protocol code APAFIS no. 3 1963-2021061014298122 v2, approved 21 July 2021).

### 2.7. Cytotoxicity Assay

To compare results between cell lines, we defined the same cell concentration for all cell lines: 1.5 × 10^3^ cells per well of 96-well plate, after optimization. Once sorted and seeded, cells were then treated for 72 h with 5-FU using a range from 0.16 to 250 µM. Three days later, the MTS reagent was added according to the manufacturer’s instructions (no. G3580, CellTiter96 AQueous One solution Cell Poliferation assay, Promega) and incubated for 3 h at 37 °C under a humidified atmosphere of 5% CO_2_. Absorbance was then measured at 490 nm with the Multiskan ™ FC 96-well plate reader (Thermo scientific ™) and the results were expressed as a percentage comparing the treated condition to the untreated condition defined as 100%. Generation of drug response curves and determination of IC50 values were performed using GraphPad Prism software (San Diego, CA, USA). We compared the obtained IC50s with those reported at https://www.cancerrxgene.org/. These experiments were also performed with oxaliplatin and irinotcan. The concentration range used for oxaliplatin was 0.13 to 200 µM and for irinotecan it was 0.4 to 650 µM. The three chemotherapies were provided by the anticancer preparation unit of the University Hospital of Limoges.

### 2.8. Proliferation Assay

Once sorted by SdFFF, cells were seeded at the same concentration as for the MTS assay, 1.5 × 10^3^ cells per well in 96-well plates. Cells were then treated for 72 h with the average IC50 values obtained for 5-FU from the cell lines and primary cultures used. Cell proliferation was assessed using the BrdU cell proliferation assay kit (no. 6813, Cell signaling technology, France) according to the manufacturer’s instructions. Absorbance was then measured at 450 nm with the Multiskan ™ FC 96-well plate reader (Thermo scientific ™, France) and the results were expressed as a ratio comparing the treated condition to the untreated condition defined as 1.

### 2.9. Cell Death Assay

From the same cell sorting by SdFFF as for the proliferation analysis, an apoptosis analysis was performed. Cell concentration and 5-FU dose and incubation time were the same as for the cell proliferation assay. Apoptosis rate was measured using the Cell Death Detection ELISA^PLUS^ kit (no. 11774425001, Roche, France), according to the manufacturer’s instructions. Absorbance was then measured at 405nm with the Multiskan ™ FC 96-well plate reader (Thermo Scientific ™) and the results were expressed as a ratio comparing the treated condition to the untreated condition defined as 1.

### 2.10. Tumorsphere Assay

Five hundred cells were seeded in nonadherent 96-well culture plates previously coated with a 10% solution of poly-2-hydroxyethylmethacrylate (no. P3932, Sigma Aldrich, France) in 95% ethanol and dried overnight at 56 °C [17]. These cells were cultured in defined medium: with serum-free GlutaMAX-DMEM/F12 (no. 10565018, Gibco ™, France) medium supplemented with 20 ng/mL epidermal growth factor (no. PHG0314, Gibco ™, France), 10 ng/mL basic fibroblast growth factor (no. PHG0264, Gibco ™, France), 0.3% glucose (no. 49163, Sigma-Aldrich, France), 20 μg/mL insulin (no. 12585-014, Gibco ™, France), 1:100 N2 supplement (no. 17502-001, Gibco ™, France) [18] and incubated at 37 °C in a humidified atmosphere of 5% CO_2_ for one week. Seven days later, spheres larger than 50 μm in diameter were counted using the Leica DMi8 microscope. Then, to study chemotherapy response in a 3D culture model, chemotherapies were added after the seven days of incubation that allow colonosphere formation, and incubated for three days. Chemotherapy doses used in the 3D chemosensitivity assays were defined from the results of the 2D cytotoxicity assays that allowed calculation of the average IC50 for all cell matrices, including cell lines and primary cultures. Colonospheres were treated using the average IC50 obtained (0.7 µM for 5-FU, 1.9 µM for oxaliplatin and 22.8 µM for irinotecan) for both single and combination therapy. Colonospheres were counted three days after treatment, imaged using the Leica DMi8 microscope, and their size was measured using ImageJ software v1.53e, National Institutes of Health

### 2.11. Statistical Analysis

All bar plots are represented by the mean ± S.E.M. of results obtained from at least three independent experiments. Flow cytometry results were represented as histograms from one representative biological replicate among the three independent experiments performed (n = 3). Significance of results was specified by stars: * *p*-value < 0.05, ** *p*-value < 0.01, *** *p*-value < 0.001. The presence of a star without a bar below indicates that the result is significantly different from the control condition: TP. Significance between the three sorted cell subpopulations was represented using the star and a bar below to allow identification of the subpopulations involved in these significantly different results. Non-significance of results is indicated by ‘ns’ for not significant, underlining that no significant differences is observed either between the sorted subpopulations or between F1/F2/F3 and the TP control. Results were analyzed by the Kruskal–Wallis test for clonogenicity assay because the results did not follow a normal distribution, verified using the Shapiro–Wilk test. Chi-squared test was used to identify the frequency of tumor-initiating cells from in vivo experiment results. The one-way ANOVA test was used for all other experiments because the results followed a normal distribution (Shapiro–Wilk test) and the comparison was between the three sorted cell subpopulations and the control. Statistical tests were performed using PAST software, version 2.17c.

## 3. Results

### 3.1. SdFFF Technique Sorts Cell Subpopulations Expressing Cancer Stem Cell Markers from Colorectal Cancer Cell Lines

First, we optimized the elution parameters—such as flow rate and field strength—for the sorting of the different cell lines used. These parameters were selected to allow an optimal separation between the peak containing the dead volume and the peak containing the cells, and are summarized in Table 2. Using a hyperlayer elution mode, cells were eluted according to their biophysical properties mainly their size and density, but also their shape and rigidity. This last parameter is deeply related to the cell differentiation status [10,12,19]. According to this elution mode, the beginning of the elution peak was more composed of large and low-density cells, while the end of the peak contains small and high-density cells [10]. Using SdFFF, we isolated three subpopulations of cells: F1, F2, and F3, as shown in the SW480 fractogram in Figure 1A. The total peak (TP), was also collected, served as a control for cell sorting efficiency, and corresponded to cells eluted from the beginning of F1 to the end of F3 (Figure 1A). Our results as our previous work showed similar results between the TP control and the not yet injected preanalytical cell sample, called crude, so the TP was used as the only control in our following experiments [10,12]. The cell size of these subpopulations was monitored by a coulter counter and confirmed that the cells eluted first (i.e., F1) are larger than those eluted last (i.e., F3) with results comparable to those published in Mélin et al. [10]. Thus, this optimization step allowed to define the elution parameters and to obtain three cell subpopulations for all the cell lines used. 

Second, we phenotypically characterized these sorted cell subpopulations by analyzing, for the first time by flow cytometry, the expression of CSC markers in the four cell lines. In the publication of Mélin et al., and until now, the analysis of CSC marker expression from SdFFF-sorted cell subpopulations was only performed by immunofluorescence, which is a less robust and objective technique than flow cytometry mainly due to the number of cells analyzed [10]. Using flow cytometry, we analyzed the expression of CD44, LGR5, BMI1, and CD133 markers within SdFFF-sorted cell subpopulations of both early-stage CRC cell lines, WiDr and SW480; and metastatic stage cell lines, SW620 and T84 (Figure 1B–F). Isotype controls were performed under the same experimental conditions to position the gates (gray line in Figure 1B). Initially, we compared the expression of these markers in the four cell lines. CSC markers were found in all cell lines but at different expression levels (Figure 1B–F). Looking at the TP values, we noticed that LGR5 and CD133 markers were more highly expressed in the early-stage cell lines while BMI1 marker expression was higher in the late-stage cell lines (Figure 1D–F). CD44 marker expression was comparable between cell lines except for the SW620 cell line that had a lower expression (Figure 1C). Thus, these results showed that these CSC markers are expressed in all four cell lines but at different levels. This distribution of expression can be explained by the original location of these cell lines. The LGR5 marker, which is one of the key markers of intestinal stem cells, was very highly expressed in both WIDR and SW480 cell lines derived from a primary patient tumor. On the other hand, the BMI1 marker—which is a marker expressed under stress by quiescent cells—was found to be predominantly expressed by the SW620 cell line, derived from lymph node metastasis, which is a transitory stage for cancer cells before they colonize another organ to develop metastases. This marker is also found in the T84 cell line, which required the tumor cells to adapt to their new environment: the lung, where the secondary metastasis developed. Thus, our results suggest that the expression of these markers is associated with the original location of these cell lines. Furthermore, interestingly, the expression levels of three of the four markers appeared to be associated with CRC stage.

Next, we focused on flow cytometry results obtained from SdFFF-sorted cell subpopulations. Figure 1B contains the histograms acquired from the three sorted subpopulations of the SW480 cell line as well as the associated isotypic control. F3 subpopulation of SW480 has a significantly higher percentage of CD44^+^ and LGR5^+^ cells compared to F1 and F2 (Figure 1B–D). For the WiDr, SW620, and T84 cell lines, the percentage of cells positive for all four markers seemed to be higher in F1 compared to the other subpopulations but without significant difference (Figure 1C–F). Therefore, our results highlight that the only significant differences observed are for the F3 subpopulation that overexpresses CD44 and LGR5 markers for SW480 (Figure 1B–D).

These phenotypic characterization results revealed two subpopulations that appear to oppose each other on the expression of CSC markers: F1 and F3. The F3 subpopulation stands out by overexpressing two CSC markers particularly important in CRC. Furthermore, we confirmed by these results that cell sorting by SdFFF is not dependent on marker expression, which offered new perspectives to study intratumoral cellular heterogeneity. Finally, phenotypic analysis provided interesting preliminary data on CSCs but needed to be completed by a functional analysis because of CSC plasticity [6,20].

### 3.2. F3 Subpopulation Has Cancer Stem Cell Functional Features in Colorectal Cancer Cell Lines

We further investigated SdFFF-sorted cell subpopulations by analyzing their functional properties. First, we examined the cell cycle distribution of the three cell subpopulations from cell lines by flow cytometry (Figure 2A,B). When comparing TP values, we noticed that the proportion of cells in G0/G1 decreased and appeared to be associated with CRC stage progression (Figure 2B). Focusing on the flow cytometry histograms of the WiDr cell line, cells in G0/G1 accounted for approximately 43% for F1, 62% for F2, and 82% for F3 (i.e., a nearly twofold difference from F1); whereas the percentage of cells in G2/M was approximately 27% for F1, 12% for F2, and 3.5% for F3 (i.e., a sevenfold difference from F1) (Figure 2A). The proportion of cells in each cell cycle phase is summarized in the bar plots (Figure 2B and Appendix A). The percentage of cells in G0/G1 was significantly higher in the F3 subpopulation compared to the other WiDr cell subpopulations (Figure 2A,B). A similar observation could be made for all cell lines; the F3 subpopulation significantly has the highest number of cells in G0/G1 phase (Figure 2B). In contrast, the percentage of cells in G2/M was significantly higher in F1 for all cell lines while the proportion of cells in S phase varied more slightly between cell subpopulations (Appendix A). Therefore, the F3 subpopulation was more quiescent or poorly proliferative compared to the other cell subpopulations for all cell lines, which is a characteristic of CSCs. Subsequently, we assessed the clonogenic capacity of these SdFFF-sorted cell subpopulations using a soft agar assay (Figure 2C,D). Comparing TP values, late-stage cell lines appeared to form slightly more colonies than early-stage cell lines (Figure 2D). The F3 subpopulation of the WiDr cell line produced significantly more colonies than F1 and TP, and a similar trend was observed for the other cell lines (Figure 2D). Interestingly, the colonies formed by the SW620 cell line, which is from the same patient as SW480 but at a more advanced stage, appeared to be more numerous and larger than those obtained by SW480 (Figure 2C). Hence, F3 was the most clonogenic subpopulation, which is also a feature of CSC.

Taken together, these functional characterization results highlighted that the F3 cell subpopulation exhibits CSC hallmarks such as clonogenicity and quiescence. By integrating the results of the phenotypic characterization, we showed that the F3 subpopulation overexpresses CSC markers and possesses CSC functional properties. In contrast, the F1 subpopulation is proliferative and poorly clonogenic. From all the characterization results of the sorted cell subpopulations, we noticed that the F2 subpopulation seemed to be an intermediate cell subpopulation between F1 and F3 with characteristics close to the TP control, so we focused the following of our study on the F1 and F3 subpopulations. However, the gold standard test for defining a CSC is its ability to initiate tumors from xenografts in immunocompromised mice [9]. Thus, the tumorigenicity of the sorted cell subpopulations remains to be determined to confirm our in vitro results.

For this purpose, we subcutaneously and for the first time inoculated the sorted F1 and F3 cell subpopulations in limiting dilution into athymic nude mice, as shown in Figure 3A. We used a single cell line for the in vivo tumor initiation assay in order to meet the guidelines of the 3R rule, and we chose the WiDr cell line based on previous results in a chick chorioallantoic membrane model [12]. The number of mice with tumors larger than 100 mm^3^ was counted seven weeks after injection (Figure 3A). Interestingly, among the 15 mice transplanted with the F1 cell subpopulation, only two mice developed tumors compared with eight mice with F3 (Figure 3B). Remarkably, F3 was the only cell subpopulation able of initiating tumor formation at the concentration of 100 injected cells (Figure 3B). Furthermore, F3 is the cell subpopulation with the highest average tumor volume, approximately 450 mm^3^ for F3 versus 360 mm^3^ for TP and 130 mm^3^ for F1, thereby a threefold difference for F3 compared to F1 (Appendix A). Importantly, a mean tumor volume above the 100 mm^3^ threshold was obtained between 36 and 39 days for F3 and TP with the 1000 cells injected condition, whereas it was only reached from 46 days for F1 (Appendix A). These results suggest that tumors obtained after injection of the F3 subpopulation are more aggressive than those obtained from F1 cells. The size of the tumors can be visualized from the photos of the tumors collected at the end of the experiment in Appendix A. Based on these in vivo results, we performed a limiting dilution analysis (LDA) using the extreme LDA software [16]. The frequency of tumor-initiating cells was estimated at one cell in 566 for the F3 subpopulation, compared with one in 3611 for F1. Thus, the frequency was four-fold higher for the F3 subpopulation compared to F1, with a significant p-value (*p* = 0.00886) (Figure 3C). Therefore, the F3 subpopulation is more tumorigenic than F1 and has the particularity to form tumors even after an injection of only 100 cells. Furthermore, F3 cells have a significantly higher frequency of tumor-initiating cells compared to F1, confirming our in vitro results and proving that the F3 subpopulation is enriched in CSCs. 

Overall, our results show that the F3 subpopulation exhibits CSC features: quiescence and clonogenicity in vitro as well as tumorigenicity in vivo and a four-fold higher frequency of tumor-initiating cells than F1 cells. This subpopulation is therefore a relevant therapeutic target, underlining the importance of studying its response to the therapies commonly used in the treatment of CRC.

### 3.3. F3 Subpopulation Is Resistant to 5-FU in Colorectal Cancer Cell Lines

We then evaluated the response to chemotherapy of the cell subpopulations sorted by SdFFF. For this purpose, we treated these subpopulations with the most commonly used chemotherapeutic molecule in CRC treatment, 5-FU, and then analyzed the induced cell viability, proliferation, and apoptosis as shown in Figure 4A. First, cells were treated with a range of 5-FU doses to obtain IC50s for each sorted cell subpopulation and cell line (Figure 4B,C). Comparing the TP values, we noticed that the T84 cell line is the most resistant to 5-FU among the four cell lines, with an IC50 of about 13 µM, which seems to underline a higher aggressiveness of this cell line (Figure 4C). IC50 values were similar between the two cell lines SW480 and SW620, and the IC50 of WiDr is approximately 5 µM, which is consistent with the results of the Genomics of Drug Sensitivity in Cancer study (Figure 4C) [21]. Then, focusing on the sorted cell subpopulations, we observed that the F3 subpopulation was significantly more resistant to 5-FU than TP and F1 for the SW480 cell line, with a similar trend for WiDr, SW620, and T84 (Figure 4B,C). Subsequently, we analyzed the impact of 5-FU treatment on cell proliferation. BrdU results showed that proliferation was significantly increased after 5-FU treatment in the F3 subpopulation for SW620 and T84 cell lines compared to F1 (Figure 4D). For both early-stage cell lines, proliferation also appeared to be higher in the F3 subpopulation compared to F1 (Figure 4D). Remarkably, F3 was the only cell subpopulation sorted by SdFFF to show increased proliferation compared to untreated conditions (dashed line). Therefore, 5-FU treatment mainly induced proliferation in the F3 cell subpopulation, whereas in the untreated condition the proliferation rate was higher in F1 (data not shown). In parallel to the proliferation analysis and using the same cell sorting, 5-FU-induced apoptosis was assessed and showed no significant change was visible between the sorted cell subpopulations for all cell lines (Figure 4E). Thus, these results highlighted the resistance of the F3 subpopulation to 5-FU. More precisely, our results showed that the enhanced viability observed in F3 is due to increased proliferation induced after treatment, without any change in the apoptosis rate (Figure 4).

We also performed these experiments after treatment with oxaliplatin and irinotecan, which are the other two chemotherapies commonly used in the treatment of CRC (Appendix A) [2,9]. The IC50 of oxaliplatin was significantly higher in the F3 subpopulation compared with F1 for the WiDr cell line, with a similar trend for SW480 and SW620 (Appendix A). Cell proliferation after treatment appeared slightly increased in the F3 subpopulation of WIDR, SW480, and SW620 compared with F1 and TP, with no significant change in apoptosis rate (Appendix A). For the T84 cell line, cell viability was comparable between the sorted cell subpopulations, whereas proliferation appeared to be slightly increased in F3 and apoptosis was significantly decreased in F1 (Appendix A). Thus, the F3 subpopulation appears to be predominantly resistant to oxaliplatin by inducing proliferation, as observed for 5-FU. Interestingly, the difference between the F1 and F3 subpopulations of T84 for proliferation and apoptosis rates was equivalent with a factor of 1.46 in both cases, showing that F1 appears to respond to treatment by resisting apoptosis while F3 resists by proliferating (Appendix A). Focusing on the results with irinotecan, we observed that the IC50 of irinotecan was significantly higher in the F3 subpopulation compared to TP for the T84 cell line, and that this trend was also noticeable for SW480 and SW620 (Appendix A). The rate of cell proliferation after irinotecan treatment appeared to be increased in the F3 subpopulation for WiDr, SW480, and T84 cell lines compared with F1 (Appendix A). Irinotecan induced apoptosis was significantly decreased in F1 and F3 compared to TP for the T84 cell line, a similar trend was observed for SW480 (Appendix A). Hence, the F3 subpopulation also appears to be mostly resistant to irinotecan, highlighted by a higher IC50 in F3 and explained by an increase in proliferation or a decrease in apoptosis.

Using SdFFF, we were able to study the individualized response of each tumor cell subpopulation to therapies. Our results showed that F3 was the most resistant of the cell subpopulations sorted to 5-FU for all cell lines (Figure 4). Similar results were obtained after treatment with oxaliplatin and irinotecan (Appendix A). Since resistance to treatment is a fundamental characteristic of CSCs, these results confirmed that the F3 subpopulation possesses CSC properties. For the first time, we identified with this label-free approach one of the mechanisms that could explain this resistance: increased cell proliferation, without any changes in the rate of apoptosis.

Our chemosensitivity experiments were performed in a two-dimensional (2D) culture model. We are aware that tumor organization is more complex and may influence the response to treatments. Thus, to overcome the limitations of this model, we further investigated the response to therapies using a three-dimensional (3D) culture model.

### 3.4. F3 Colonospheres Are Resistant to Single and Combination Chemotherapies in Colorectal Cancer Cell Lines

We then examined the impact of 5-FU-based chemotherapies on the colonospheres from the SdFFF-sorted cell subpopulations. To this end, we performed the sphere formation assay, a 3D cell culture model mimicking tumor organization in vitro. This test relies on the ability of cancer cells to form colonospheres—i.e., microtumor-like spheroids—from a single progenitor cell and is used to assess the tumorigenic potential of solid tumors in vitro [6]. Thus, we developed a protocol using our label-free approach as illustrated in Figure 5A. First, all four cell lines have the ability to form colonospheres (Figure 5B). However, colonosphere morphology and size varied between cell lines, with colonosphere size appearing to correlate with the stage of CRC development (Figure 5B). After 5-FU treatment, the number of colonospheres was significantly increased in the F3 subpopulation compared with F1 for both advanced-stage lines, with a similar trend for SW480 (Figure 5C). Colonospheres were also more abundant in the F3 subpopulation than in TP for WiDr after oxaliplatin treatment, with similar observations for SW480 and SW620 (Appendix A). After irinotecan treatment, F3 was the most resistant of the cell subpopulations for SW480, with a similar trend for SW620 and T84 (Appendix A). Chemotherapy is most often administered in combination to patients with late-stage CRC, but sometimes also to early-stage patients with risk factors for recurrence [9], in order to potentiate the anticancer effect and prevent recurrence. Accordingly, we evaluated the response of sorted cell subpopulations to chemotherapy combinations—such as FOLFOX, FOLFIRI, and FOLFIRINOX—in addition to the response analysis to single chemotherapies. Combinations of 5-FU-based chemotherapies mainly induced an increase in colonospheres in the F3 subpopulation, with a significant difference between F3 and TP and/or F1 for WiDr after FOLFOX and FOLFIRINOX treatments, as well as for WiDr, SW480, and T84 after FOLFIRI (Figure 5D–F). Thus, our results underline a chemoresistance of the F3 subpopulation to both monochemotherapy and combination chemotherapy.

Overall, these results confirmed those obtained in 2D. The F3 subpopulation was predominantly the most resistant to monochemotherapy and chemotherapy combinations among the sorted cell subpopulations. As observed in 2D, the results in 3D indicate that F3 cells resist by inducing proliferation after treatment, revealed by an increased ability to form colonospheres and a larger size. Thus, in our study, we identified two therapeutically relevant cell subpopulations: the F1 subpopulation appears to be chemosensitive while the F3 is chemoresistant, from both early and advanced CRC cell lines. Our findings demonstrated that our SdFFF-based cell sorting approach is capable of revealing chemoresistance in a cell subpopulation. Nevertheless, we continued our analysis to test our approach on primary cultures that are more representative of tumor heterogeneity and complexity. 

### 3.5. F3 Subpopulation Exhibits CSC-like Features and Chemoresistance in Primary Colorectal Cancer Cultures

To confirm the promising results obtained from the cell lines, we performed the same experiments using CRC primary cultures (CPP). CPP14 was established from an early-stage CRC while CPP35 was derived from a tumor that had invaded the peritoneum—i.e., at a more advanced stage—with clinical data summarized in Table 1. As previously implemented on the cell lines, the cell subpopulations sorted by SdFFF were characterized at the phenotypic and functional levels. Phenotypically, both primary cultures expressed markers of CSC. However, higher levels of the markers LGR5 and BMI1 were observed in CPP35 (tumor-invaded peritoneum) compared to CPP14 (early-stage tumor), which may be explained as for the cell lines by the location of the original tumor (Appendix A). CPP35 has reached all layers of the colonic wall where a strong expression of LGR5 is found and also the peritoneum, which corresponds to a pre-metastatic stage with a strong cellular stress that can be associated with the expression of BMI1. Interestingly, we noticed the same pattern of expression of CD44 and BMI1 markers between CPP14 and early-stage cell lines, and of CD44, BMI1, and CD133 markers between CPP35 and late-stage cell lines (Appendix A). In contrast, no significant differences in marker expression were observed between the sorted cell subpopulations (Appendix A). Next, we investigated the cell cycle distribution of these two primary cultures. G0/G1 cells were significantly more abundant in F3 compared to the other subpopulations for CPP35, with a similar trend for CPP14 (Appendix A). Conversely, the number of cells in G2/M was significantly lower in F3 for both primary cultures (Appendix A). These results highlight a relative quiescence of the F3 subpopulation, as observed in the cell lines. Afterwards, clonogenicity results showed that the F3 subpopulation forms significantly more and larger colonies than F1 for CPP14, with a similar trend for CPP35 (Appendix A). Thus, our characterization findings highlight CSC characteristics for F3 including clonogenicity and quiescence as observed in the cell lines. Among the three cell subpopulations, F1 and F3 stood out from a functional point of view, and therefore we focused on these two subpopulations for chemosensitivity assays. As outlined for the cell lines, we demonstrated the ability of SdFFF to sort cell subpopulations from primary cultures.

Next, we evaluated the response of these sorted cell subpopulations to chemotherapies in 2D and 3D cell culture models for each primary culture. In 2D, we analyzed cell viability after 5-FU treatment, as well as induced proliferation and apoptosis (Figure 6A–C). No significant difference was observed between the IC50 values of the cell subpopulations (Figure 6A). After 5-FU treatment, the proliferation rate appeared to be higher in the F3 subpopulation of CPP35 whereas 5-FU induced apoptosis was significantly higher in the F1 subpopulation for both primary cultures (Figure 6B,C). Thus, the F1 subpopulation seemed to be more sensitive to 5-FU due to its susceptibility to apoptosis compared to TP and F3 in both primary cultures (Figure 6C). For oxaliplatin and irinotecan, the F3 subpopulation has an IC50 that appears very slightly higher for both primary cultures, significantly higher proliferation for CPP35, and an apoptosis rate that seems lower than F1 (Appendix A). Thus, our 2D results highlight a chemoresistance of the F3 subpopulation predominantly in both primary cultures and mainly related to a better resistance to apoptosis compared to F1.

Finally, we investigated the response of SdFFF-sorted cell subpopulations in a 3D culture model to 5-FU alone or in combination with other chemotherapies. In the absence of treatment, both primary cultures had the ability to form colonospheres (Figure 6D). Quantitatively, CPP14 formed more colonospheres than CPP35 without treatment, 18 and 2.5 colonospheres respectively when comparing TP values (Figure 6E). Remarkably, F3 generated significantly twice as many colonospheres as the other cell subpopulations for CPP14 (Figure 6E). Under treated condition, the F3 subpopulation of CPP35 formed significantly more colonospheres than F1 after 5-FU treatment, with a similar trend after oxaliplatin and irinotecan treatment, and chemotherapy combinations (Figure 6F–I and Appendix A). The significant difference in untreated condition between F1 and F3 of CPP14 could explain the minor differences observed in treated condition (Figure 6E–I). The colonosphere results from the primary cultures showed that F3 is predominantly the most chemotherapy-resistant subpopulation of SdFFF-sorted cells, mainly due to resistance to apoptosis compared to F1 (Figure 6). Thus, the results obtained from the primary cultures mirrored those obtained from the cell lines. In both cases, the F3 subpopulation emerged from the functional characterization results. F3 cells are clonogenic, quiescent, and chemoresistant to single and combined therapies, which relates them to CSCs. Furthermore, the ability of these cells to initiate tumors in vivo, which is the gold standard, was confirmed.

Therefore, our study highlighted the relevance of our label-free approach to study the individualized response of each tumor cell subpopulation to chemotherapies from cell lines and primary cultures in order to identify chemoresistances.

## 4. Discussion

Our study addressed the impact of intratumoral cellular heterogeneity in the context of treatment resistance, without relying on surface marker expression to sort cell subpopulations. Regardless of cell line and stage, the SdFFF technique allowed isolation of cell subpopulations with distinct phenotypical and functional characteristics. Compared to previous results [10], we showed here the presence of two therapeutically relevant cell subpopulations based on their CSCs characteristics: F1 and F3. This was achieved by in vivo tumor initiation assay and in vitro chemosensitivity assays. The F1 subpopulation was proliferative and chemosensitive, while the F3 subpopulation exhibited CSC functional hallmarks, including the ability to initiate tumors in mice and chemoresistance. These two cell subpopulations of interest were both identified from early- and late-stage cell lines. From the primary cultures, we also identified the F3 subpopulation as a CSC-enriched fraction due to its clonogenicity and quiescence. Thus, the results of the functional characterizations obtained with the primary cultures mirrored those obtained for the cell lines. Furthermore, we demonstrated the chemoresistance of the F3 CSC-like subpopulation in the two primary cultures, as observed in the cell lines. However, the mechanisms developed by F3 cells to resist chemotherapies are different between cell lines and primary cultures. Indeed, the F3 subpopulation of cell lines induce increased proliferation after 5-FU treatment without modifying apoptosis, while F3 from primary cultures does not significantly change its proliferation rate but decreases its apoptosis rate compared to F1. Therefore, thanks to our label-free approach, we highlighted a cell subpopulation with a CSC-like phenotype that may play a crucial role in tumor progression and metastasis. Furthermore, we were able to identify different mechanisms used by these cells to resist chemotherapies.

Initially identified in acute myeloid leukemia, CSCs were later discovered in many solid cancers such as CRC based on the expression of surface markers [22]. In CRC, early publications on CSCs used surface markers as a prerequisite to identify these cells [23,24,25,26,27,28,29]. Many CSC markers have been identified in CRC and are reviewed in Hervieu et al. [9]. Our phenotypic characterization results were initially disappointing due to the lack of significant differences between the cell subpopulations sorted for three of the four cell lines studied, although a trend did emerge. However, several publications have questioned the use of these markers since they are also expressed by intestinal stem cells and cancer cells [20,30,31]. Furthermore, Prasetyanti and Medema point out that CSC markers can be considered as a highly context-dependent property of cells [32]. Furthermore, the use of surface markers is hampered by the plasticity of CSCs, which is another obstacle to their identification and isolation. Shimokawa et al. and de Sousa e Melo et al. showed that ablation of LGR5^+^ CSCs limits tumor growth but does not prevent tumor recurrence due to re-emergence of LGR5^+^ CSCs from proliferating LGR5^-^ cells [33,34]. Intriguingly, another study demonstrated that dissemination and metastatic colonization were carried out by LGR5^-^ cells in CRC, with subsequent re-emergence of LGR5^+^ CSCs at the metastatic site [7]. Thus, the results of these publications suggest that LGR5^-^ non-CSCs are able to transform into LGR5-expressing CSCs, highlighting the phenotypic plasticity of these cells that appears to be crucial for primary tumor growth and metastasis. These results highlight the complexity of identifying as well as isolating these CSCs due to their shared expression with non-CSCs and their cellular plasticity. All of these data converge on evidence that stemness is not as hierarchical and fixed as originally thought, but rather dynamic and endowed with considerable plasticity. It is this polydispersity of CSCs that justified the SdFFF label-free cell sorting approach, offering new perspectives to isolate CSC-enriched subpopulations.

A consensus has emerged suggesting that their functional capabilities, particularly their tumorigenic potential and chemoresistance, are more reliable for identifying CSCs than surface markers [35]. Our results showed that cells with the ability to initiate tumors in mice and chemoresistance were not necessarily correlated with cells expressing CSC markers. The publication of Lenos et al. is in agreement with our observations, demonstrating that there is a divergence between cells positive for CSC markers and cells with CSC functionality [36]. In our study, the results of the in vivo tumor initiation assay provide evidence that F3 is a subpopulation of CSCs with high tumorigenicity (8 out of 15 tumor-bearing mice) as well as a high frequency of tumor-initiating cells compared to F1. Nevertheless, F1 cells also possess tumor-initiating capacity, but much less efficiently than F3 cells, with only 2 out of 15 mice developing a tumor. These results as well as the results of the characterization of the sorted cell subpopulations suggest that F3 cells are the CSCs at the top of the cell hierarchy, whereas F1 cells have already started to differentiate and lose their CSC characteristics. F1 cells can be considered as transit-amplifying cells that are known to be highly proliferative, which is consistent with our cell cycle results, and engaged in a differentiation process that explains the maintenance of some CSC characteristics but much less efficient than F3. Accumulating evidence suggests that cell–cell interactions and crosstalk within the tumor microenvironment (TME) can modulate cellular state, stemness, and many fundamental characteristics of CSC [32,37,38]. Our in vivo results suggested cooperation of non-CSC cancer cells with CSCs. Indeed, we noticed that for the TP control, five mice carried tumors while the CSC-enriched subpopulation had only four, which may indicate that the presence of both F3 cells (CSCs) and F1 cells (non-CSCs) in TP promoted tumor development. As F3 cells are CSCs, when injected alone, these cells are able to initiate a tumor and promote its growth through their ability to differentiate multi-lineage in order to recreate tumor heterogeneity. For F1 cells already engaged in a differentiation process, tumor formation is significantly less efficient and takes longer to reach the threshold tumor volume of 100 mm^3^, approximately 46 days for F1 versus 37 days for F3. Interestingly, for TP, the threshold volume is reached very slightly before the F3 subpopulation, which demonstrates the importance of cellular heterogeneity in tumor development with the requirement of CSCs for tumor initiation, as well as the presence of non-CSCs to enhance tumor growth.

CSCs are of particularly therapeutic interest. Although chemotherapies kill most tumor cells, CSCs are able to escape the lethal effect of these drugs, which can lead to tumor recurrence. One of the main reasons for treatment failure is that anticancer drugs often only target actively cycling tumor cells and therefore do not affect CSCs, which are frequently in a quiescent or minimally proliferative state. The results of our study support this hypothesis: the cell subpopulation enriched in CSC features was quiescent/low proliferative and chemoresistant, while the subpopulation of actively cycling cells was more sensitive. Similar results have been reported in Kreso et al. [39]. Remarkably, our study showed that only the CSC-enriched cell subpopulation escaped chemotherapy by significantly increasing treatment-induced cell proliferation without any change in cell death, demonstrating cell plasticity. These 2D results show that the chemoresistance of F3-CSCs corresponds to the exit of their quiescent state and the transition to a proliferative state, allowing the reconstitution of cellular heterogeneity. This is also illustrated by our results in the 3D model of the WiDr cell line in response to 5-FU-based chemotherapies. Furthermore, our results highlight that resistance and therapeutic escape are a functional property of CSCs, reinforcing the relevance of our approach based on label-free cell sorting by SdFFF. Collectively, our in vitro results reflect what frequently occurs in CRC patients: chemotherapies kill proliferative cells (i.e., the F1 subpopulation), resulting in tumor regression; but fail to target CSCs (i.e., the F3 subpopulation), which are resistant and evade therapy, resulting in cancer relapse in patients.

Therapies targeting CSCs are a promising therapeutic approach. However, the development of anticancer agents capable of specifically targeting CSCs has proven very difficult or has shown limited efficacy [40]. These disappointing results may be explained by the fact that the study models used often fail to reproduce the heterogeneous tumor architecture of patients [32]. One of our original hypotheses was that cell subpopulations isolated from early-stage CRC cell lines might behave differently to chemotherapies than those from metastatic stages. Although one of the two metastatic cell lines had the highest IC50 for each of the chemotherapies tested, the results were not as pronounced as initially expected. Cancer cell lines grown in 2D have traditionally been used to model cancer, but the absence of the microenvironment may impact response to treatment [32]. Cellular heterogeneity and plasticity may also compromise treatment efficacy, as highlighted in the studies of Shimokawa et al. and de Sousa e Melo et al. [32,33,34,38]. CRC patient-derived cultures, such as primary cultures, better model the heterogeneity and complexity of patient tumors, which are essential to consider when studying response to therapy. In our study, the IC50 differences are more pronounced between the two different stage primary cultures, especially for oxaliplatin and irinotecan, which may provide insights for more personalized therapy in patients from whom the primary cultures were derived. In addition to optimizing the biological model used, 3D cell culture increases the sensitivity of chemotherapy response studies [32]. Our 3D chemosensitivity results from microtumor-like spheroids confirmed and refined those obtained in 2D. Importantly, the CSC-enriched subpopulation sorted by SdFFF has the ability to survive in suspension in serum-free media and proliferate even in the presence of chemotherapy. The transition from a quiescent state under physiological conditions to a proliferative state after stress may suggest a dormant CSC state. Indeed, F3 CSCs leave the cell cycle and become quiescent/dormant (G0/G1 arrest), allowing them to escape chemotherapies targeting highly proliferative cells, while retaining the potential to reiterate proliferative expansion [41]. Thus, cell proliferation plays a crucial role for this subpopulation in response to stress such as chemotherapies. Therefore, our study models based on label-free cell sorting provide new insights to study responses to therapies and resistance mechanisms developed by CSCs. Nevertheless, key components of TEM must be incorporated into our model to mimic patient tumors [32]. Future cancer therapies will have to take into account both the CSCs and non-CSCs that form the tumor mass, as well as the surrounding TEM.

In summary, we have demonstrated the relevance of using SdFFF to obtain a chemoresistance signature. Thus, the CSC-enriched subpopulation will allow to specifically explore the signaling pathways associated with therapeutic resistance. Since this technique provides a model reflecting intratumoral cellular heterogeneity, it will be valued in the study of the response to therapies of each cell subpopulation of a biological sample. Our study thus opens new perspectives for the understanding of CSC-related resistance and offer solutions specifically adapted to a more personalized and targeted therapy of colorectal cancer.

## Figures and Tables

**Figure 1 cells-11-02264-f001:**
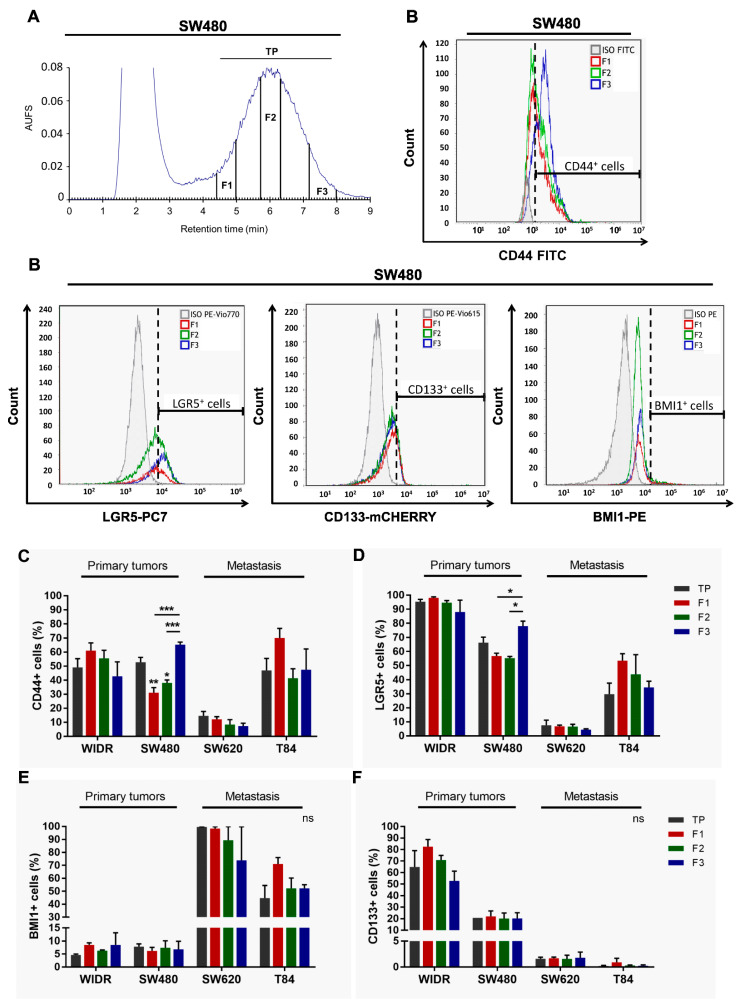
Phenotypic characterization of SdFFF sorted cell subpopulations from CRC cell lines. (**A**) Fractogram of the SW480 cell line obtained by SdFFF with the sorted cell subpopulations: total peak control (TP), F1, F2, and F3. AUFS: absorbance units full scale. (**B**) The expression level of CSC markers, CD44, LGR5, BMI1, and CD133, was assessed by flow cytometry and plotted as histograms for the SW480 cell line. Graphs show one representative biological replicate (n = 3). (**C**–**F**) Quantification of CD44 (**C**), LGR5 (**D**), BMI1 (**E**), and CD133 (**F**) positive cells was summarized in the bar plot for all cell lines. All these results are represented as means ± SEM and statistical differences with ns means a non-significant result both between the sorted cell subpopulations and between a sorted subpopulation and the control, * *p*-value < 0.05, ** *p*-value < 0.01, *** *p*-value < 0.001, and * alone for significant results compared to TP using one-way ANOVA test.

**Figure 2 cells-11-02264-f002:**
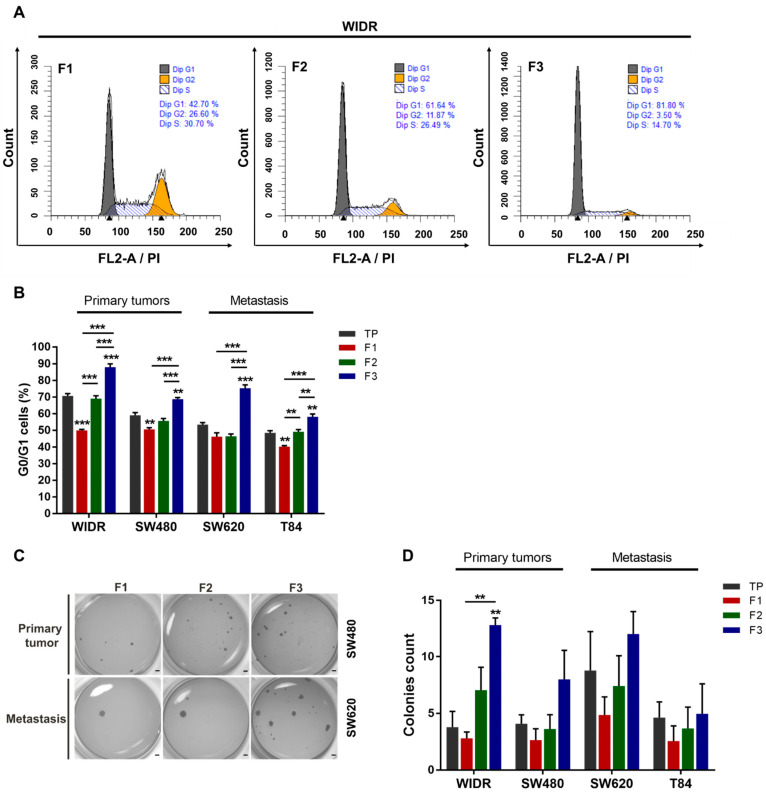
Functional characterization of SdFFF-sorted cell subpopulations from CRC cell lines highlighting a quiescent and clonogenic F3 subpopulation. (**A**) The distribution of the sorted cell subpopulations in the cell cycle was analyzed by flow cytometry and plotted as histograms using modfit software. Graphs show one representative biological replicate (n = 3). (**B**) Quantification of G0/G1 cells was performed for each sorted cell subpopulation and cell line, and presented in the bar plot from three biological replicates. (**C**,**D**) Cell clonogenicity was assessed by a soft agar assay. The results were presented as images of the wells with formed colonies (**C**) as well as the quantification of the colony count in the bar plot (**D**). Scale bar, 1 mm. All these results are represented as means ± SEM and statistical differences with ** *p*-value < 0.01, *** *p*-value < 0.001, and ** alone for significant results compared to TP using one-way ANOVA test for cell cycle distribution analysis and Kruskal–Wallis test for clonogenicity.

**Figure 3 cells-11-02264-f003:**
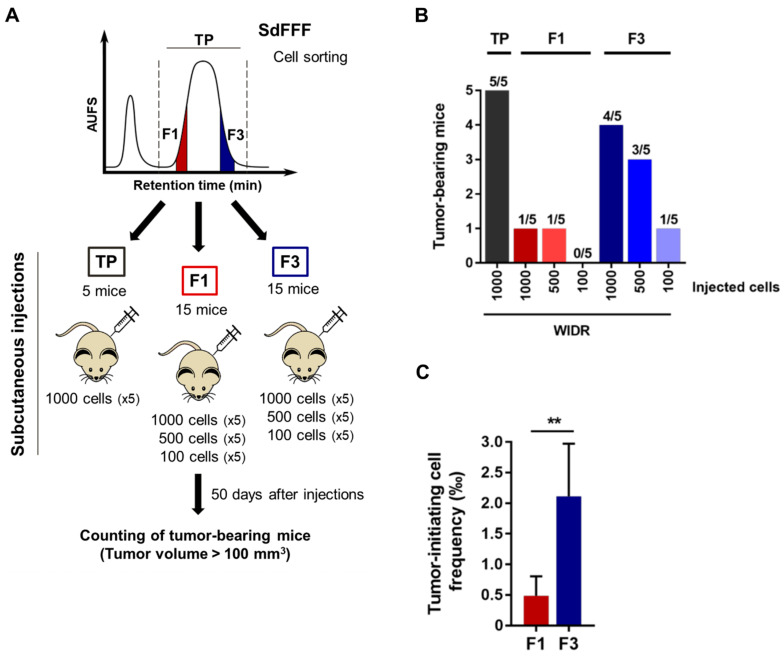
Tumorigenic capacity of the F3 subpopulation in immunodeficient mice, even after an injection of only 100 cells. (**A**) Schematic representation of the in vivo tumor initiation assay. Nude mice were injected subcutaneously with either 1000, 500, or 100 F1 and F3 cells of the WiDr cell line. (**B**) The number of tumor bearing mice (tumor > 100 mm^3^) was evaluated seven weeks after injection and summarized in the bar plot. (**C**) Tumor initiating cell frequency was determined from in vivo results using extreme limiting dilution analysis (ELDA) software. Results are represented as means ± SEM and statistical differences with ** *p* value < 0.01 using chi-squared test of ELDA software (https://bioinf.wehi.edu.au/software/elda/, accessed on 28 January 2022).

**Figure 4 cells-11-02264-f004:**
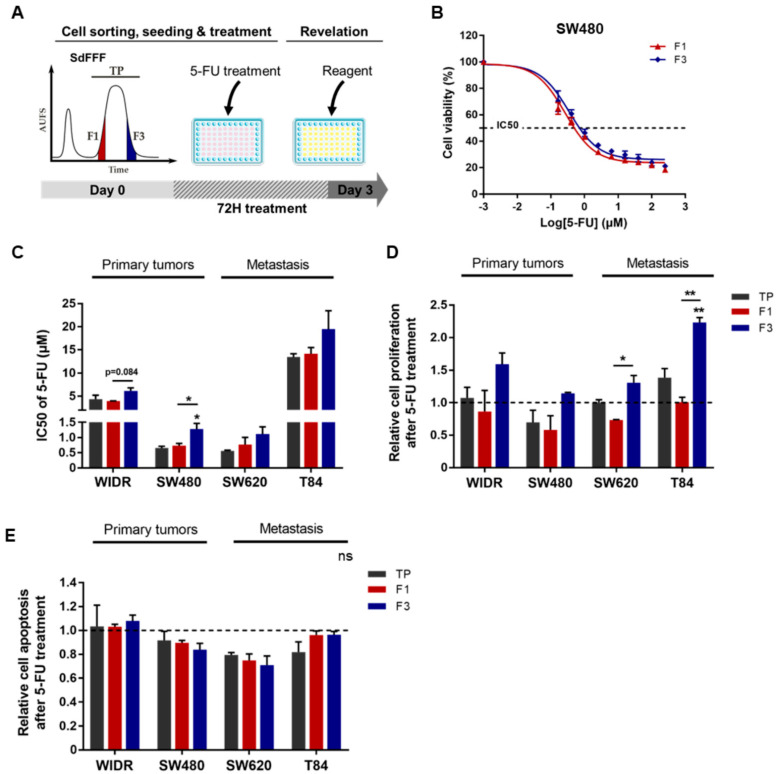
5-FU resistance of F3 subpopulation from CRC cell lines. (**A**) Schematic representation of 5-FU response analysis performed. After cell sorting by SdFFF, cell subpopulations were treated with 5-FU for three days and then cell viability, proliferation, and apoptosis analyses were performed. (**B**,**C**) After 5-FU treatment, cell viability analyzed by MTT assay was presented as a dose–response curve for the SW480 cell line (**B**) and a bar plot with IC50 values obtained from all cell lines (**C**). (**D**) Cell proliferation rate after 5-FU treatment was measured by BrdU assay and presented in the bar plot as a ratio between treated and untreated conditions (dashed line). (**E**) After 5-FU treatment, apoptosis rate was measured using ELISA cell death assay and compared with the untreated condition (dashed line). All these results are represented as means ± SEM and statistical differences with ‘ns’ means a non-significant result both between the sorted cell subpopulations and between a sorted subpopulation and the control, * *p*-value < 0.05, ** *p*-value < 0.01, and * alone for significant results compared to TP using one-way ANOVA test.

**Figure 5 cells-11-02264-f005:**
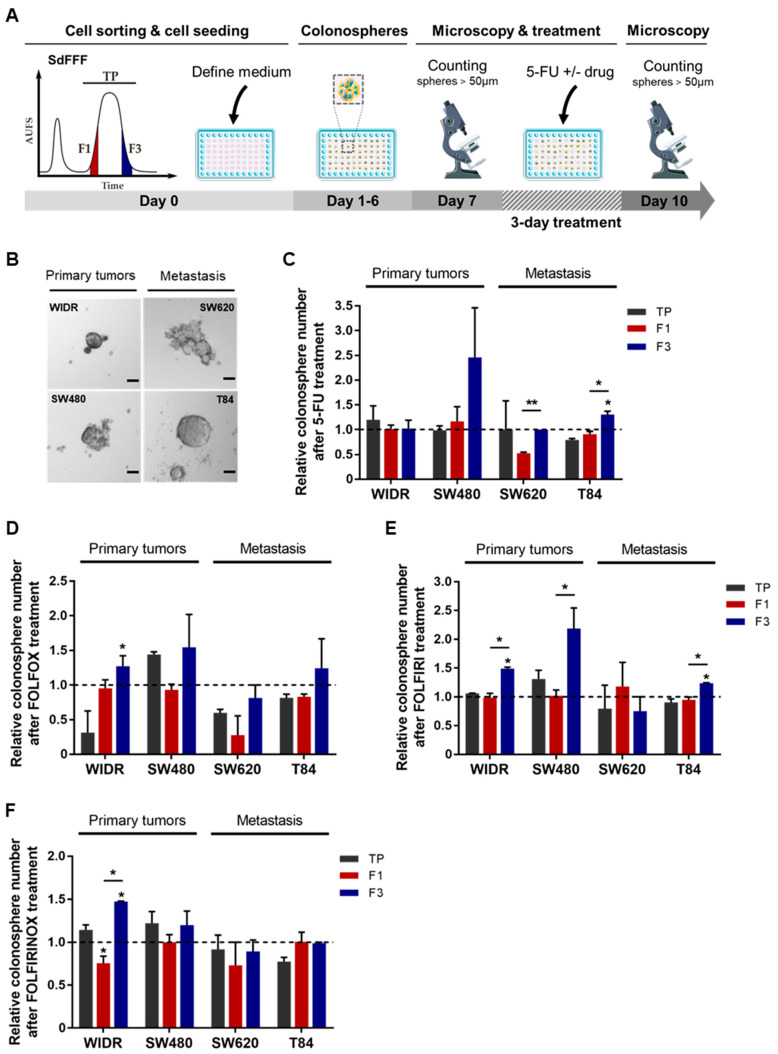
Chemoresistance of F3 colonospheres to 5-FU alone or combined with oxaliplatin and/or irinotecan from CRC cell lines. (**A**) Schematic representation of chemotherapy response analysis performed from colonospheres. SdFFF sorted cell subpopulations are seeded at low density in defined medium for seven days and then treated with chemotherapy for three days. Colonospheres with a diameter larger than 50µM were counted before and after treatment. (**B**) The in vitro tumorigenic potential of cells was assessed by a sphere formation assay. Scale bar 50 µm. (**C**–**F**) The impact of chemotherapies on CSC tumorigenic properties was analyzed in vitro after treatment with 5-FU alone (**C**), or combined with either oxaliplatin (FOLFOX) (**D**), irinotecan (FOLFIRI) (**E**), or both (FOLFIRINOX) (**F**), compared with the untreated condition (dashed line). All these results are represented as means ± SEM and statistical differences with * *p*-value < 0.05, ** *p*-value < 0.01, and * alone for significant results compared to TP using one-way ANOVA test.

**Figure 6 cells-11-02264-f006:**
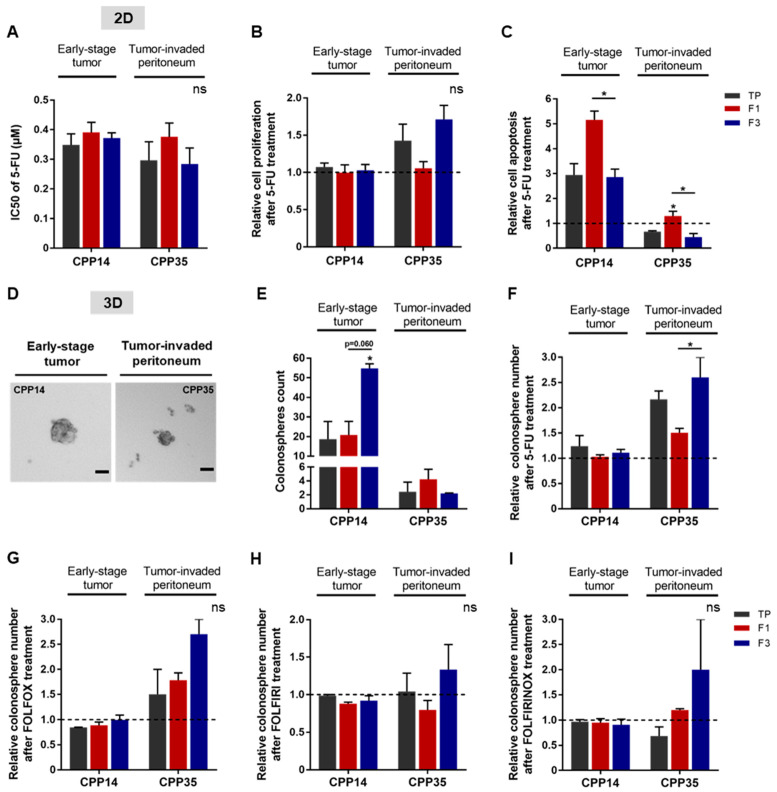
Chemoresistance of F3 subpopulation from both primary cultures. (**A**) After 5-FU treatment, cell viability was analyzed by MTT assay and presented as a bar plot with IC50 values obtained from all primary cultures. (**B**) Post-treatment induced proliferation was assessed by BrdU assay, comparing the results to the untreated condition (dashed line). (**C**) The apoptosis rate of both primary cultures was analyzed by ELISA cell death assay. (**D**,**E**) The tumorigenic potential of cells in vitro was evaluated by a sphere formation assay. In untreated condition, the ability of primary cultures to form colonospheres was presented as images of formed colonospheres (**D**) and their quantification summarized in the bar plot (**E**). Scale bar 50 µm. (**F**–**I**) The impact of chemotherapies on CSC tumorigenic properties in vitro was investigated after treatment with 5-FU alone (**F**) or in combination: FOLFOX (**G**), FOLFIRI (**H**), or FOLFIRINOX (**I**). All these results are represented as means ± SEM and statistical differences with ns means a non-significant result both between the sorted cell subpopulations and between a sorted subpopulation and the control, * *p*-value < 0.05 and * alone for significant results compared to TP using one-way ANOVA test.

**Table 1 cells-11-02264-t001:** Clinical data from colorectal cancer patients from whom primary cultures were established.

Primary Cultures	CPP14	CPP35
pTNM	T2N0M0 Early-stage tumor	T4aN0M0 Tumor-invaded peritoneum
Stages	Stage I	Stage IIb
Primary location	Left colon	Transverse colon
Chemotherapy	No	No
Radiotherapy	No	No
Curative surgery	Yes	Yes

**Table 2 cells-11-02264-t002:** Optimal elution condition for cell sorting of CRC cell lines and primary cultures by SdFFF.

Cell Lines and Primary Cultures	Cell Concentrations (Cells/mL)	Field (g)	Flow Rate (mL/min)
WiDr	2 × 10^6^	8	0.8
SW480	2.5 × 10^6^	8	0.8
SW620	2.5 × 10^6^	8	0.8
T84	3 × 10^6^	15	0.8
CPP14	2.5 × 10^6^	10	0.8
CPP35	2.5 × 10^6^	8	1

## Data Availability

UMR INSERM 1308-CAPTuR “Control of Cell Activation in Tumor Progression and Therapeutic Resistance” lab.

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
