# Peer review of "A Label-Free Cell Sorting Approach to Highlight the Impact of Intratumoral Cellular Heterogeneity and Cancer Stem Cells on Response to Therapies"

_cells, 2022, doi:10.3390/cells11152264_

Round 1

Reviewer 1 Report

The manuscript "A label-free cell sorting approach to highlight the impact of intratumoral cellular heterogeneity and cancer stem cells on response to therapies" reported the results of the analysis of cell subpopulations of colorectal cancer cell lines and primary cultures using the sedimentation field-flow fractionation technique (SdFFF) to separate according to their biophysical properties such as size, density, shape, and rigidity. This study demonstrates that the one (F3) of the three sorted cell subpopulations exhibits a higher frequency of tumor-initiating cells, proved by higher clonogenicity in vitro and tumorigenicity in vivo. This subpopulation also demonstrated higher quiescence and chemoresistance compared to the other two subpopulations. Therefore, the authors considered that these features indicate their cancer stem cell potential. The authors offered the SdFFF approach to isolate CSC-enriched subpopulations and study the individualized response of sorted tumor cell subpopulations to develop personalized resistance-treating therapy.

This work provides an advance toward an understanding of the intra-tumor cellular heterogeneity and cancer stem identification. The novelty and significance are sufficiently substantiated in the Introduction and Discussion. The cell assays, xenograft analysis, and statistical data processing were carried out accurately, with all required controls. The Results description is detailed. The list of references is reasonably sufficient and includes all relevant publications. In general, the conclusions are supported by the results. In the Discussion section, the authors briefly review the results of previous studies and compare them with their own data. In my opinion, the present work may interest a broad audience.

In general, the results are described carefully, but several issues need clarification. I have the following comments:

1.  1. In experiments on the study of tumorigenicity in vivo, differences in the frequency of tumor formation between the studied subpopulations were found. As shown in Figure 3B, the incidence of tumor formation upon inoculation of 1000 cells in the TP control subpopulation is 100% (5/5) vs. 20% (1/5) for F1 and vs. 80% (4/5) for F3. In the Discussion, the authors explain the differences in tumor formation rates between TP and F3 and the ability of F1 cells to initiate tumors in 20% of mice (Fig 3B, C) by the cooperation between CSCs and non-CSCs and various TME factors that mediate cancer plasticity. Therefore, in a TP population containing approximately 20% F3-CSCs, the tumorigenicity was significantly enhanced after their interaction F3-CSCs with non-CSC F1 and F2, even more than most enriched F3-CSCs?

2. It can be assumed that the heterogeneity of all subpopulations increases with time. However, how can the F1 and F2 populations be characterized and classified in terms of their ability to initiate a tumor (the gold standard)? Are these cell populations transitive or mixed? Also, the authors' ideas about how transitions between states can differ are unclear. What direction dominates - the direction of strengthening the CSC or non-CSC state? These questions require clarification in the Discussion.

3. The results of chemoresistance experiments should show all significant differences between the subpopulations compared to untreated controls, and between TP, F1, and F3 populations, in Figures 4, 5, and 6. Which groups does the designation ns refer to graphs in Figure 6? This is important for understanding the complete picture of chemoresistance of the studied cell subpopulations in primary tumors and metastases cultivated in 2D and 3D culture systems, as well as the scientific soundness of the conclusions.

4. A more specific explanation of the F3 subpopulation chemoresistance is needed (L669-672 and L699-701):

"Remarkably, our study showed that only the CSC-enriched cell subpopulation escaped chemotherapy by significantly increasing treatment-induced cell proliferation without any change in cell death, demonstrating cell plasticity." Does this mean that F3-CSC chemoresistance is the exit of their quiescent state and the transition to a non-CSCs proliferative state after drug exposure?

"Importantly, the CSC-enriched subpopulation sorted by SdFFF has the ability to survive in suspension in serum-free media and proliferate even in the presence of chemotherapy." How does their proliferation correlate with a dormant CSC state? 

Author Response

Reviewer 1 :

1) In experiments on the study of tumorigenicity in vivo, differences in the frequency of tumor formation between the studied subpopulations were found. As shown in Figure 3B, the incidence of tumor formation upon inoculation of 1000 cells in the TP control subpopulation is 100% (5/5) vs. 20% (1/5) for F1 and vs. 80% (4/5) for F3. In the Discussion, the authors explain the differences in tumor formation rates between TP and F3 and the ability of F1 cells to initiate tumors in 20% of mice (Fig 3B, C) by the cooperation between CSCs and non-CSCs and various TME factors that mediate cancer plasticity. Therefore, in a TP population containing approximately 20% F3-CSCs, the tumorigenicity was significantly enhanced after their interaction F3-CSCs with non-CSC F1 and F2, even more than most enriched F3-CSCs?

In Figure 3, the injected F3 cells formed tumors in 8 out of 15 mice compared to 2 out of 15 mice for the F1 cells, which shows that the F3 subpopulation is more tumorigenic than F1 (Figure 3B). At the quantity of 1000 injected cells, four mice carried tumors in F3 versus one mouse for F1 and five for the control condition. In the group of mice injected with 1000 F3 cells, two mice had to be euthanized before the end of the experiment. The reasons for early euthanasia were for the first one the reaching of the maximum tumor volume of 1000 mm3, and for the second one, carrying a tumor larger than 600 mm3, a deterioration of its general health as well as a significant weight loss, underlining the aggressiveness of these two tumors. Remarkably, F3 cells were the only ones to form a tumor at the amount of only 100 injected cells, proving the stemness of these cells and their ability to initiate tumors in limiting dilution (Figure 3B). The TP results (5 out of 5 mice with tumor) suggest that the tumor-initiating cells in the TP control population are F3 cells but that the cooperation and interaction of F3 cells (CSCs) with F1 cells (non-CSCs), both of which are present in the TP population, promote tumor development. However, the tumor volume reached at the end of this experiment by the F3 subpopulation is higher compared to the TP control, highlighting an improved tumor growth and progression for the F3 CSC compared to F1 and TP (Figure S1D and E). Changes were made in the discussion to further detail and discuss our in vivo results: L802-819 and L826-834.

2) It can be assumed that the heterogeneity of all subpopulations increases with time. However, how can the F1 and F2 populations be characterized and classified in terms of their ability to initiate a tumor (the gold standard)? Are these cell populations transitive or mixed? Also, the authors' ideas about how transitions between states can differ are unclear. What direction dominates - the direction of strengthening the CSC or non-CSC state? These questions require clarification in the Discussion.

The gold standard for defining a CSC is in vivo xenotransplantation into immunodeficient mice. We did not analyze the tumorigenicity of the F2 subpopulation with the mouse model. From the results of the phenotypic and functional characterization of the cell lines, we have shown that the F2 subpopulation seems to be an intermediate population between F1 and F3 with characteristics close to the TP control. Thus, we chose to focus on the F1 and F3 subpopulations for the in vivo experiment, in order to limit the number of mice used and to meet the 3R rule, as well as the chemotherapy response tests. As mentioned in the response to the previous comment (1), our results show a difference in tumorigenicity between F1 and F3. Indeed, F3 cells have a higher tumorigenic capacity compared to F1 with respectively an incidence of tumor formation of 80% vs. 20% at 1000 injected cells, 60% vs. 20% at 500 injected cells and 20% vs. 0% at 100 injected cells. Using the ELDA assay, the frequency of tumor-initiating cells was estimated to be 1 in 3611 cells for F1 versus 1 in 566 cells for F3, highlighting a fourfold higher frequency in F3 compared to F1 (Figure 3C). Thus, the in vivo results provide evidence that F3 is a subpopulation of CSCs, with high tumorigenicity as well as a high frequency of tumor initiating cells. To a lesser extent, F1 cells retain an ability to initiate tumors, but much less efficiently than F3 cells. These results suggest that F3 cells are the CSCs at the very top of the cell hierarchy while F1 cells have already started to differentiate and lose CSC characteristics. F1 cells can be considered as transit-amplifying cells that are highly proliferative, as demonstrated for F1 cells in the cell cycle analysis, and engaged in a differentiation process which explains the maintenance of some CSC characteristics but much less efficient. According to our results, the dominant transition is from the CSC state to the non-CSC state. These subpopulations are therefore transitive rather than mixed. As suggested, changes have been made in the discussion to clarify the points raised in this comment: L812-819, L826-834 and L876-880.

3) The results of chemoresistance experiments should show all significant differences between the subpopulations compared to untreated controls, and between TP, F1, and F3 populations, in Figures 4, 5, and 6. Which groups does the designation ns refer to graphs in Figure 6? This is important for understanding the complete picture of chemoresistance of the studied cell subpopulations in primary tumors and metastases cultivated in 2D and 3D culture systems, as well as the scientific soundness of the conclusions.

We thank the reviewer for this comment, which allow us to clarify our figures. Significant differences between subpopulations F1, F2 and F3 are indicated by the presence of stars (p-value) above a bar connecting them. The presence of a star just above the histogram (without bar) refers to a significant difference between the sorted subpopulation and the PT control. Finally, the presence of "ns" for "not significant" in a graph means a non-significant result both between the sorted cell subpopulations as well as between a sorted subpopulation and the control. The only significant difference not shown in our figures is between the different cell lines to avoid overrepresentation. Our results were analyzed and presented in this manner throughout our manuscript, as specified in the materials and methods section and in the figure legend. Following your comment and in order to clarify this in our article, we have made changes to lines 250 to 253: “underlining that no significant differences is observed either between the sorted subpopulations or between F1/F2/F3 and the TP control” as well as lines L323, L563 and L713: “ns means a non-significant result both between the sorted cell subpopulations and between a sorted subpopulation and the control”.

4) A more specific explanation of the F3 subpopulation chemoresistance is needed (L669-672 and L699-701):

"Remarkably, our study showed that only the CSC-enriched cell subpopulation escaped chemotherapy by significantly increasing treatment-induced cell proliferation without any change in cell death, demonstrating cell plasticity." Does this mean that F3-CSC chemoresistance is the exit of their quiescent state and the transition to a non-CSCs proliferative state after drug exposure?

Indeed, under treatment, F3-CSCs resist chemotherapy mainly by emerging from their quiescent state and increasing their proliferation rate. The transition to a proliferative state is correlated with a non-CSC state. However, the transition between a quiescent and proliferative state is transient and dependent on environmental cues. In this case, it is conceivable that post-treatment induced proliferation aims to compensate for cell loss and reform cell heterogeneity. In the absence of stress, the cells can then become quiescent again. In response to this comment, a few changes were made to lines L785-789 and L845-847.

"Importantly, the CSC-enriched subpopulation sorted by SdFFF has the ability to survive in suspension in serum-free media and proliferate even in the presence of chemotherapy." How does their proliferation correlate with a dormant CSC state?

The reviewer's comment is extremely relevant and points to a question we have as well. Our explanation is as follow: our study allows us to visualize the behavior of each cell subpopulation to chemotherapies commonly used in the clinic. In the absence of therapies, the CSC-enriched subpopulation is quiescent and is able to form colonospheres due to their ability to self-renew. In the presence of chemotherapies and in both 2D and 3D models, the cells will exit their quiescence and resist by proliferating. Indeed, the results observed in our study may suggest a dormant CSC state in which F3-CSCs exit the cell cycle and become dormant (G0/G1 arrest), allowing them to escape chemotherapies targeting highly proliferative cells, while retaining the potential to reiterate proliferative expansion (Phan, T.G., Croucher, P.I. The dormant cancer cell life cycle. Nat Rev Cancer 20, 398–411 (2020). https://doi.org/10.1038/s41568-020-0263-0). In this publication, the dormant state is associated with cell cycle arrest in G0/G1 which may coincide with the results observed in our cell cycle analysis. Following your comment, we have added text in the discussion of line 876 to 880.

Reviewer 2 Report

Comments to the Author

In this manuscript, Hervieu et.al. used sedimentation field-flow fractionation (SdFFF) technique to sort the colorectal cancer (CRC) cell lines and acquired cell subpopulations featuring distinct phenotypic and functional characterizations related to the tumorigenicity and cancer stem cells. The authors further quantitively analyzed the chemoresistance of different cell subpopulations using 5-FU and 5-FU-based chemotherapies. The results demonstrated the effect of intertumoral heterogeneity on the chemoresistance of CRCs. In my view, this is an interesting study that combined the non-invasive sorting of tumor cells with chemoresistance studies. Although lack of mechanism-based in-depth studies, however, this manuscript still provides useful information that may interest the readers who are interested in the studies of tumor cellular heterogeneity and personalized medicines. Therefore, I recommend publication, but some minor points need to be clarified.

1.    As I understand, all four tumors tested in this study are stable tumor cell lines. The label “Primary tumors” seems a bit misleading. Were these tumors from primary patient samples?

2.    Figure 1B and S1A, the flow cytometry curves should be presented in the same plot.

3.    In Figure 3. The authors concluded that F3 subpopulation was more tumorigenic than F1 and F2; therefore, the tumorigenic capacity of F3 should be higher than that of TP. Would you please clarity the reason that the mice from TP group achieved the biggest tumor volume?

4.    In the chemoresistance study, it is unclear that how the authors determine the dose schedule of each drug in the combined therapies.

5.    Please revise the description of figure legends. I do not suggest including result descriptions in the figure legends.

6.    The resolution of all figures should be improved. The current version is blurry.

Author Response

Reviewer 2:

1) As I understand, all four tumors tested in this study are stable tumor cell lines. The label “Primary tumors” seems a bit misleading. Were these tumors from primary patient samples?

In our overall study, we used four cell lines and two primary cultures, all from patients. For the cell lines, we used two cell lines established from a primary tumor of a patient with early stage CRC (WIDR and SW480) as well as two cell lines established from CRC metastasis: lymph node (SW620) and lung (T84). The "primary tumor" and "metastasis" labels indicated in the graphs allow to specify the location of origin of these cell lines and to compare early versus advanced stages of CRC. This information is detailed in the materials and methods. For the two primary cultures used in this study, the first was established from a primary CRC tumor at an early stage (T2N0M0) while the second was established from a tumor that has already invaded the peritoneum and is therefore at a more advanced stage but without lymph node invasion or metastasis (T4aN0M0). The information on the primary cultures is specified in the material and method and in Table 1.

To clarify this point, we have specified the labels used in the graphs in the Materials and Methods to avoid confusion. We have added for the cell lines "labeled ‘primary tumors’ in the graphs" to line L95, and "as well as the two metastatic cell lines, labeled ‘metastasis’ in the graphs" to line L96. For the primary cultures, we also made changes in the text, adding “labeled ‘early-stage tumor’ in the graphs" to line L111 and "whereas CPP35, labeled ‘tumor-invaded peritoneum’ in the graphs, was" to L112-113.

2) Figure 1B and S1A, the flow cytometry curves should be presented in the same plot.

The authors would like to thank the reviewer for this pertinent comment. We have corrected this in the new version of the manuscript by including all three sorted cell subpopulations as well as the isotypic control in the same graph (Figure 1B).

3) In Figure 3. The authors concluded that F3 subpopulation was more tumorigenic than F1 and F2; therefore, the tumorigenic capacity of F3 should be higher than that of TP. Would you please clarity the reason that the mice from TP group achieved the biggest tumor volume?

In Figure 3, the injected F3 cells formed tumors in 8 out of 15 mice compared to 2 out of 15 mice for the F1 cells, which shows that the F3 subpopulation is more tumorigenic than F1 (Figure 3B). At the quantity of 1000 injected cells, four mice carried tumors in F3 versus one mouse for F1 and five for the control condition. In the group of mice injected with 1000 F3 cells, two mice had to be euthanized before the end of the experiment. The reasons for early euthanasia were for the first one the reaching of the maximum tumor volume of 1000 mm3, and for the second one, carrying a tumor larger than 600 mm3, a deterioration of its general health as well as a significant weight loss, underlining the aggressiveness of these two tumors. Remarkably, F3 cells were the only ones to form a tumor at the amount of only 100 injected cells, proving the stemness of these cells and their ability to initiate tumors in limiting dilution (Figure 3B). The TP results (5 out of 5 mice with tumor) suggest that the tumor-initiating cells in the TP control population are F3 cells but that the cooperation and interaction of F3 cells (CSCs) with F1 cells (non-CSCs), both of which are present in the TP population, promote tumor development.

As F3 cells are CSCs, when injected alone, these cells are able to initiate a tumor and promote its growth through their ability to differentiate multi-lineage in order to recreate tumor heterogeneity. For F1 cells already engaged in a differentiation process, tumor formation is significantly less efficient and takes longer to reach the threshold tumor volume of 100mm3, approximately 46 days for F1 versus 37 days for F3 (Figure S1E). Interestingly, for TP, the threshold volume is reached very slightly before the F3 subpopulation, around 36 days post-injection, which demonstrates the importance of cellular heterogeneity in tumor development with the requirement of CSCs for tumor initiation as well as the presence of non-CSCs to enhance tumor growth. However, the tumor volume reached at the end of this experiment by the F3 subpopulation is higher compared to the TP control, highlighting an improved tumor growth and progression for the F3 CSC compared to F1 and TP (Figure S1D and E). Changes were made in the discussion to further detail and discuss our in vivo results: L802-819 and L826-834.

4) In the chemoresistance study, it is unclear that how the authors determine the dose schedule of each drug in the combined therapies.

The doses of the chemotherapies used in the 3D chemosensitivity tests were based on the results of the 2D tests. In our 2D cytotoxicity assay, we used wide ranges of chemotherapy concentrations: for 5-FU from 0.16 to 250µM, for oxaliplatin from 0.13 to 200µM and for irinotecan from 0.4 to 650µM, to obtain IC50s for each cell line and primary culture, as summarized in the table below. The concentrations of the ranges were optimized and defined in order to use the same ranges for all cell lines and primary cultures and to obtain the full dose-response curve as presented in Figure 4B. From these results, we calculated the average of the IC50 obtained for all the cell matrices tested (4 cell lines + 2 primary cultures), summarized in the table below. When treating with a combined therapy, the average IC50 of one chemotherapy was added to the average IC50 of the other chemotherapy. Thus, when we performed the 3D chemosensitivity tests, we treated the formed colonospheres with the IC50 averages determined in 2D. Following your comment, these indications have been added in the material and method on lines L234 to 238.

Chemotherapy

Concentration ranges used in 2D

Average IC50 obtained for all cell lines and primary cultures

Minimum concentration

Maximum concentration

5-FU

0.16 µM

250 µM

0.7 µM

Oxaliplatin (OXA)

0.13 µM

200 µM

1.9 µM

Irinotecan (IRI)

0.4 µM

650 µM

22.8 µM

FOLFOX

(5-FU + OXA)

5-FU 0.7 µM
+ OXA 1.9 µM

FOLFIRI

(5-FU + IRI)

5-FU 0.7 µM
+ IRI 22.8 µM

FOLFIRINOX

(5-FU+OXA+IRI)

5-FU 0.7 µM
+ OXA 1.9 µM
+ IRI 22.8 µM

5) Please revise the description of figure legends. I do not suggest including result descriptions in the figure legends.

As suggested in your comment, we have simplified the legends of our figures by removing the description of the results. Thus, changes have been made to lines L317, L320, L404, L407, L411, L455, L457, L555, L559, L562, L605, L610, L695, L698, L701, L705, L707 and L710. Similar changes were made in the legends of the supplementary figures.

6) The resolution of all figures should be improved. The current version is blurry.

We thank the reviewer for this comment. We have made some changes in the figures, notably in the size of the graphs and the font used to improve the readability of the figures (main text and supplementary data). These modifications are present in the new version of the manuscript sent with our answers to the reviewers' remarks.

Reviewer 3 Report

This manuscript seeks to study the role of intra-tumoral cellular heterogeneity in the evolution of cancer stem cells and chemo-resistance in established colon carcinoma-derived cell lines and patient derived primary cells. This study demonstrates an innovative application of label-free sedimentation field flow fractionation technique and effective utilization to isolate and characterize putative cancer initiating stem cell population.

   Overall, this well-written manuscript contains appropriate descriptive methodologies, systematic analyses of the data generated, and appropriate data interpretation. The discussion section is well-supported with relevant published evidence.

   On the other side, the Results section is rather descriptive and therefore, compromises the clarity of the data generated. This limitation can be easily remedied by revising the results section to enhance the clarity and scientific impact of interesting data.

1.      Results: This section in its present form is overtly descriptive. Revised version that adequately emphasizes the scientific significance of the data generated is required to enhance the clarity.

2.      Discussion: Data from the primary cultures isolated from colon cancer patients represents an important aspect that provides a link for the preclinical data to their clinical translatability. The data from established colon cancer cell lines and from patient-derived cells need to be compared for their common and unique characteristics. This aspect needs to be appropriately emphasized in the Discussion section.

Author Response

Reviewer 3:

1) Results: This section in its present form is overtly descriptive. Revised version that adequately emphasizes the scientific significance of the data generated is required to enhance the clarity.

We would like to thank the reviewer for this relevant comment. We have made changes to the results section to improve the clarity of our data and manuscript. The changes involve the lines: L297-311, L335-345, L361-365, L369-373, L381-399, L419-424, L436-438, L444-449, L492-500, L505-509, L512-513, L523-525, L574-577, L584-588, L597-598, L615-634, L642-650, L657-659, L661-670, L680-691 and L727-742.

2) Discussion: Data from the primary cultures isolated from colon cancer patients represents an important aspect that provides a link for the preclinical data to their clinical translatability. The data from established colon cancer cell lines and from patient-derived cells need to be compared for their common and unique characteristics. This aspect needs to be appropriately emphasized in the Discussion section.

We have taken the reviewer's comment into consideration and have made changes in the discussion section. These changes can be found on lines L754-768.

Round 2

Reviewer 1 Report

In the revised manuscript, the authors clarified their viewpoints and statistical evaluation. They responded to all reviewers' comments. The revised version has been significantly improved.